# LEARNING COMMUNICATION-EFFICIENT OPTIMIZERS

## ABSTRACT

Communication-efficient variants of SGD, specifically local SGD, have received a great deal of interest in recent years. These approaches compute multiple gradient steps locally, that is on each worker, before averaging model parameters, helping relieve the critical communication bottleneck in distributed deep learning training. Although many variants of these approaches have been proposed, they can sometimes lag behind state-of-the-art optimizers for deep learning. In this work, we incorporate local optimizers that compute multiple updates into a learned optimization framework, allowing to meta-learn potentially more efficient local SGD algorithms. Our results demonstrate that local learned optimizers can substantially outperform local SGD and its sophisticated variants while maintaining their communication efficiency. We show that the learned optimizers can generalize to new datasets and architectures, demonstrating the potential of learned optimizers for improving communication-efficient distributed learning.

## 1 INTRODUCTION

Rapidly training large-scale deep learning models is a problem of continued interest in the community. It requires a great deal of distributed computing resources that are often challenging to efficiently utilize. In many distributed learning settings, the communication overhead associated with distributed SGD can often lead to inefficient use of computing resources and reduced wall clock times (Lin et al., 2018). This reliance on frequent communication is especially impractical for training large models over heterogeneous hardware (Yuan et al., 2022). Moreover, it can increase the cost and complexity of designing data center and other infrastructure to support the heavy communication constraints.

The primary communication overhead of distributed SGD comes from the synchronization of gradients computed by different workers. A recently popular direction to alleviate this overhead is local SGD (Stich, 2019), where each worker computes multiple gradient steps independently before aggregating the weights of their local models. This reduces the communication costs.

Local SGD, however, has a number of challenges limiting its practical use. Firstly, as the number of local steps increases the local models may diverge from each other leading to a degradation of performance (Wang et al., 2019). Secondly, local SGD can also be challenging to combine with state-of-the-art adaptive optimization strategies that are needed to efficiently train many model classes such as transformers (Vaswani et al., 2017). Finally, local SGD introduces a complex dynamic between the local and global updates, which can for example lead to complex interactions between hyperparameters such as global and local learning rates (Reddi et al., 2020).

Learned optimization through meta-learning has been an increasingly important topic of research interest (Andrychowicz et al., 2016). Advances have been made in both scalable architectures (Wichrowska et al., 2017) and meta-learning strategies (Vicol et al., 2021). Recent works have demonstrated highly competitive performance with state-of-the-art adaptive optimization strategies (Metz et al., 2022a;b). Notably, Metz et al. (2022b) showed that scaling these approaches by increasing the variety of meta-training scenarios and sophistication of the architecture can lead to strong meta-generalization on new architectures and datasets. This progress suggests that these approaches can potentially serve as off-the-shelf replacements for existing adaptive optimization methods.

In this work, we propose learned optimization as an approach to alleviate the challenges of communication-efficient distributed learning. We, therefore, take the first steps to investigate if

learned optimization can be used to improve communication-efficient distributed learning, and in particular local SGD and its variants. Our main contributions are:

- We demonstrate that learned optimizers can be used to augment local SGD for communication-efficient distributed deep learning, outperforming strong baselines and maintaining benefits when the number of local steps increases.
- We propose and evaluate two architectures for the learned optimization of local SGD.
- We demonstrate that our local learned optimizers, even when meta-learned on a single or few architecture and dataset combinations, can generalize to new architectures and datasets obtaining competitive results in communication-efficient distributed settings.

Overall, our results outline a promising future for communication-efficient distributed learning. We will release our code publicly upon publication.

## 2 RELATED WORK

**Local SGD and communication-efficient DL**  Local SGD has been analyzed in a number of works (Stich, 2018; Lin et al., 2018) which demonstrated that it both theoretically and empirically can lead to communication savings. It has also been shown that local SGD, particularly when combined with phases of regular SGD, can lead to better generalization (Lin et al., 2018), while Ortiz et al. (2021) found this generalization trend is less clear at large scale.

Wang et al. (2019) introduced the use of global or server-side momentum and showed that it can accelerate local SGD as well as a number of decentralized and asynchronous stochastic algorithms. A closely related algorithm has been proposed and extensively used in federated learning for communication efficiency (McMahan et al., 2017; Li et al., 2019). Work in this field has largely focused on addressing the heterogeneity of data across workers or clients (Karimireddy et al., 2020; Mishchenko et al., 2022). These advancements are generally achieved by hand-designed algorithmic enhancements, whereas our approach relies on more flexible and potentially more powerful learnable mechanisms that may generalize these and more complex algorithms.

Another approach to communication-efficient learning is to compress the gradients or parameters. Two popular strategies in this setting are sparsification (Stich et al., 2018; Shi et al., 2019) and quantization (Alistarh et al., 2017) of the gradient. These strategies have also been combined in Wang et al. (2023). This line of work is thus orthogonal but complementary to our proposal. Communication efficiency has also been studied in the decentralized setting (Nabli & Oyallon, 2022; Nabli et al., 2023; Lian et al., 2018). Our work focuses on the centralized training setting but the methods can also be extended to decentralized training.

**Learning to Optimize (L2O)**  The idea of learning to learn and meta-learning has a long history (Schmidhuber, 1992; Thrun & Pratt, 2012). Many early works in this area focused on learning to efficiently acquire general knowledge or inductive bias. Hochreiter et al. (2001) proposed to use meta-learning in direct combination with gradient-based optimization to learn a separate network, which can be seen as a learned optimizer, which performs updates on another network. Andrychowicz et al. (2016) extended these ideas to a more scalable LSTM-based per-parameter architecture and demonstrated that the learned optimizer can generalize to new problems.

A large number of follow up works have improved L2O methods (Wichrowska et al., 2017; Metz et al., 2019; Chen et al., 2020; Metz et al., 2020; Harrison et al., 2022), see Chen et al. (2022); Amos (2022) for surveys. These methods introduced different types of hierarchy into the learnable optimizer while simplifying its architecture in favor of stronger predefined features to improve its efficiency. In Metz et al. (2022a), the efficiency of these methods was further analyzed in a large-scale study and a highly efficient and simple per-parameter MLP model and feature extraction approach was introduced, which we leverage in our work. However, compared to our work these have not considered a distributed setting, where learnable optimizers may significantly alleviate the communication bottleneck.

Ji et al. (2019) proposed to learn the aggregation of gradients from workers in a distributed learning framework with a recurrent network. However, the focus was on improving non-local SGD while

our work focuses on the communication efficiency in settings where each worker returns a message computed from multiple update steps. Furthermore, our approach is shown to generalize to new architectures and datasets.

## 3 METHODOLOGY

Our method builds upon the local SGD framework (Stich, 2019), by learning to aggregate local model weights $\{\boldsymbol{w}_{t,h}^{(k)}\}_{k=0}^{K-1}$ during communication rounds. Specifically, at each communication round $t$, on all $K$ clients, we take $H$ local step of SGD using a local minibatch of size $B_{loc}$ for each local step $h$. After $H$ local steps, we employ a per-parameter learned optimizer $F_\phi$ to compute the updated centralized weights. $F_\phi$ receives as input the difference between the initial and final weights ($\Delta_t^{(k)}$) for each worker $k$ and the learned optimizer state ($\boldsymbol{u}_t$); it outputs the global update. We provide a detailed description of the process in Algorithm 1. By computing the centralized update using an expressive neural network $F_\phi$, our method can be seen as a generalization of existing update methods such as taking the average iterate (Stich, 2018) or computing server-side momentum updates (Wang et al., 2019).

---

**Algorithm 1:** Learned Local Optimization

**Data:** Number of iterations $T$; Number of workers $K$; Number of local steps $H$; Local learning rate $\gamma$; Initial weights $\boldsymbol{w}_{0,0}$; Initial learned optimizer state $\boldsymbol{u}_0$; Dataset $\mathcal{D}$; Loss function $\mathcal{L}$; Learned optimizer $F_\phi$

1 **for** $t \in \{0, 1, \dots, T-1\}$ **do**
2    **for** $k \in \{0, 1, \dots, K-1\}$ ***in parallel*** **do**
3      **for** $h \in \{0, 1, \dots, H-1\}$ **do**
4        $X_h^{(k)}, Y_h^{(k)} \leftarrow \text{GET\_LOCAL\_MINIBATCH}(\mathcal{D})$
5        Local step: $\boldsymbol{w}_{t,h+1}^{(k)} \leftarrow \boldsymbol{w}_{t,h}^{(k)} - \gamma \nabla_{\boldsymbol{w}} \mathcal{L}\left(X_h^{(k)}, Y_h^{(k)}; \boldsymbol{w}_{t,h}^{(k)}\right)$
6      Difference in weights after $H$ local steps: $\Delta_t^{(k)} \leftarrow \boldsymbol{w}_{t,H}^{(k)} - \boldsymbol{w}_{t,0}^{(k)}$
7    $\Delta_t = \frac{1}{K} \sum \Delta_t^{(k)}$
8    Compute AdaFactor features and update state: $\mathbf{A}_t, \boldsymbol{u}_{t+1} = \text{ADA}(\boldsymbol{w}_{t,0}, \boldsymbol{u}_t, \Delta_t)$
9    Global update: $\boldsymbol{w}_{t+1,0} \leftarrow F_\phi\left(\mathbf{A}_t, \Delta_t^{(0,1,\dots,K-1)}\right)$

---

### 3.1 LEARNED OPTIMIZER TRAINING AND ARCHITECTURES

We consider the meta-learning framework with a learned optimizer $F_\phi$ with parameters $\phi$ that is used to optimize a model with parameters $\boldsymbol{w}$. In the meta-learning formulation, $\phi$ is obtained by solving the following optimization problem:

$$\min_\phi \mathbb{E}_{(\mathcal{D}, \boldsymbol{w}_0) \sim \mathcal{T}} \mathbb{E}_{(X,Y) \sim \mathcal{D}} \left( \frac{1}{T} \sum_{t=0}^{T-1} \mathcal{L}(X, Y; F_\phi(\cdot)) \right), \tag{1}$$

where $\mathcal{T}$ is a distribution over optimization tasks defined as pairs of dataset $\mathcal{D}$ and initial weights $\boldsymbol{w}_0$ associated with a particular neural architecture, $\phi$ represents the weights of the learned optimizer, and $T$ is the length of the unroll which we write as a fixed quantity for simplicity. In practice, during meta-optimization, we can vary $T$ according to a truncation schedule (Metz et al., 2022a).

In our experiments, $F_\phi$ is an MLP with 2 hidden layers and 32 hidden nodes per layer. The input to $F_\phi$ is based on a diverse set of features computed based on $\Delta_t$ and state $\boldsymbol{u}_t$ such as different kinds of momentum analogous to adaptive optimizers and AdaFactor features (Shazeer & Stern, 2018; Metz et al., 2022a). Their computation is detailed in supplement A.

**LOpt-A** Our first proposed variant of a locally learned optimizer uses $\Delta_t$, the average of the updates from all workers, as an input feature and uses it to compute features along with the optimizer state. This process is analogous to existing learned optimization proposed in Metz et al. (2022a) where the role of the gradient is replaced with $\Delta_t$.

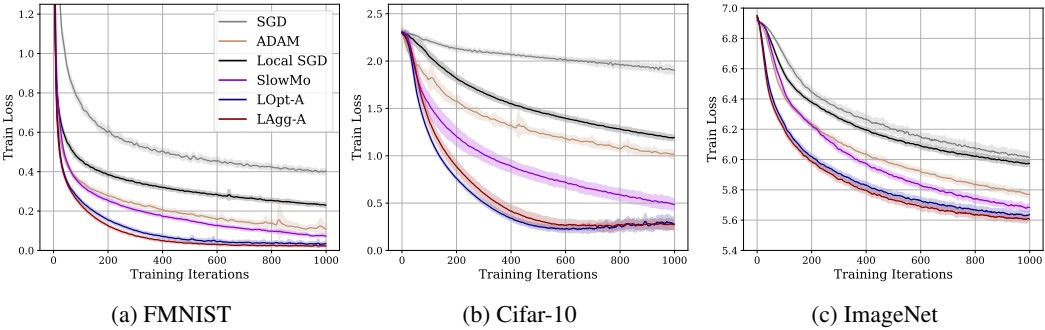

(a) FMNIST          (b) Cifar-10          (c) ImageNet

Figure 1: **Learned optimizers enable communication-efficient learning.** Our LOpt-A and LAgg-A outperform strong communication-efficient baselines such as SlowMo and local SGD. They also outperform well tuned standard optimization strategies at equivalent effective batch sizes.

**LAgg-A**   Our second locally learned optimizer takes advantage of pre-aggregated information from each worker, specifically it uses all the $\Delta_t^{(k)}$ as input to the MLP along with the AdaFactor features computed from $\Delta_t$, the average of the updates from all workers: $F_\phi\left(\mathbf{A}_t, \Delta_t^{(0,1,\dots,K-1)}\right)$. We refer to it as a *locally learned aggregator* as it learns to aggregate the weights updates coming from $K$ workers. This variant generalizes our LOpt-A and is potentially more powerful, however, we found that LOpt-A can also perform well while being simpler.

As discussed in Reddi et al. (2020) the class of local algorithms can be described with a server-side optimizer and worker-side optimizer. For example, SlowMo (Wang et al., 2019) can be interpreted as adding momentum to the server optimization. Our design of the learned optimizer architecture only parameterizes the server-side optimization making its use more practical and scalable. Specifically, standard learned optimizers have an overhead of memory and compute. The memory must store state information and intermediate activations of the learned optimizer. In the case of our learned optimizer, this overhead (Metz et al., 2022a) is only incurred at the aggregation stage. Similarly, while the computational cost of the forward pass of learned optimizers provides a substantial overhead compared to simple add and multiply operations of SGD and Adam, for the case of our learned optimizer this cost becomes small with respect to the large amount of data processed on workers during local updates.

## 4   EXPERIMENTS

Our empirical evaluation is based on standard supervised learning tasks with different dataset and architecture combinations commonly studied in learned optimization literature Metz et al. (2022a). All of the datasets and architectures in our study are presented here. For each task, we use a local batch size $B_{loc}$ of 128, while the rest of the configuration varies depending on the experiment.

We use the Fashion MNIST dataset (10 classes) with full-size $28 \times 28$ images with 1 channel which we refer to as FMNIST or FMNIST $28 \times 28$. We also use the CIFAR-10 dataset (10 classes) with full-size $32 \times 32$ images with 3 channels, referred this dataset as CIFAR-10 or CIFAR-10 $32 \times 32$. Finally, we use the ImageNet dataset (1000 classes) with downsampled size $32 \times 32$ images with 3 channels. We refer to this dataset as ImageNet or ImageNet $32 \times 32$.

As for neural network architectures, we use multilayer perceptron (MLP) of two different sizes, both with ReLU activations. The first has two layers of 128 hidden nodes each and we refer to it as 2-Layer MLP. The second has three hidden layers of 128 hidden nodes each and we refer to it as 3-Layer MLP. We also use a convolutional neural network (CNN) of 3 layers with ReLU activations. All 3 layers have convolution kernels of size $3 \times 3$ and use same padding. The first layer has 32 units and uses size 2 stride while the two other layers have 64 units and use size 1 stride. We refer to this architecture as CNN. The number of output values depends of the dataset with which the architecture is used.

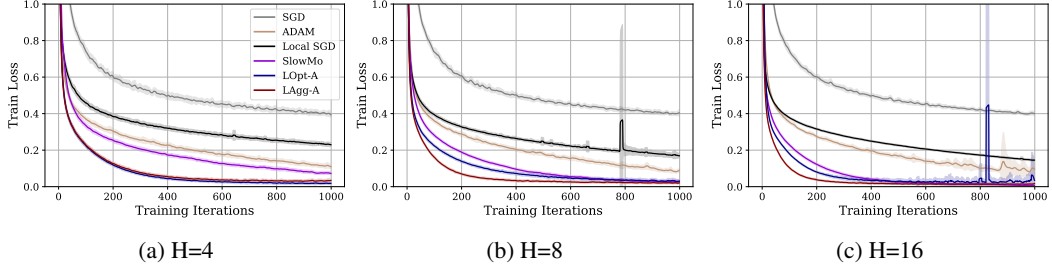

(a) H=4                          (b) H=8                          (c) H=16

Figure 2: **LAgg-A outperforms all optimizers for** $H \in \{4, 8, 16\}$ **local steps**. All training curves are reported for the $28 \times 28$ FMNST dataset. The top row plots training curves for a small CNN, while the bottom row plots training curves for an MLP. All experiments use $K = 8$.

Table 1: **Communication rounds until acheiving** $0.2$ **loss value for different optimizers at different H values.**

| Optimizer | H=4 | H=8 | H=16 |
|-----------|-----|-----|------|
| Local SGD | – | 721 | 625 |
| SlowMo | 311 | 182 | 121 |
| LOpt-A | **119** | 121 | 89 |
| LAgg-A | 122 | **81** | **55** |

In our experiments, we first establish the superior convergence of our learned optimizers when evaluated in-distribution (Figure 1). We also demonstrate that our method scales to larger values of H (Figure 2), larger values of K (Figure 3), and illustrate the importance of AdaFactor features to our model's success (Figure 5). We evaluate the performance of our learned optimizers on test loss when targeting the validation loss during meta-training (Figure 4). Finally, we evaluate the generalization capabilities of our models demonstrating they can obtain strong performance on unseen datasets and unseen architectures (Figure 6) as well as generalize to different settings of H (Figure 7).

## 4.1 EXPERIMENTAL DETAILS

The following subsection provides a brief overview of the meta-training process of our learned optimizers and presents the baseline optimizers used within our study.

**Meta-training LOpt-A and LAgg-A**    To meta-train our learned optimizers we estimate gradients using Persistent Evolutionary Strategies (PES) (Vicol et al., 2021) and take gradient descent steps using AdamW and a linear warmup plus cosine decay schedule. Each gradient is estimated from a batch of 8 tasks each unrolled to a specific number of steps, $N$. $N$ varies throughout training according to a log-uniform truncation schedule with minimum and maximum values of $N = 100$ steps and $N = 1000$ steps, respectively. Throughout our experiments, gradients are estimated with respect to the optimizee's training loss, except for the curves in Figure 4 whose gradients were estimated with respect to the optimizee's validation loss. During meta-training, the learning rate is warmed up for 100 steps to a maximum learning rate before being decayed (following a cosine decay schedule) to $1/3$ of the maximum value. Extensive meta-training details are provided in the supplement B.

**Baselines**    To provide a comparison to non-local algorithms, we train models using **SGD** (Robbins, 1951) and **Adam** (Kingma & Ba, 2017) for a number of steps equivalent to the total number of communication rounds used for the local methods. At each step, these baselines compute updates using the same effective batch size $K \times H \times B_{loc}$ as the local optimizers they are compared to. The hyperparameters for SGD and Adam are provided in the supplement C. For each setting, we provide the best-performing hyperparameter combination.

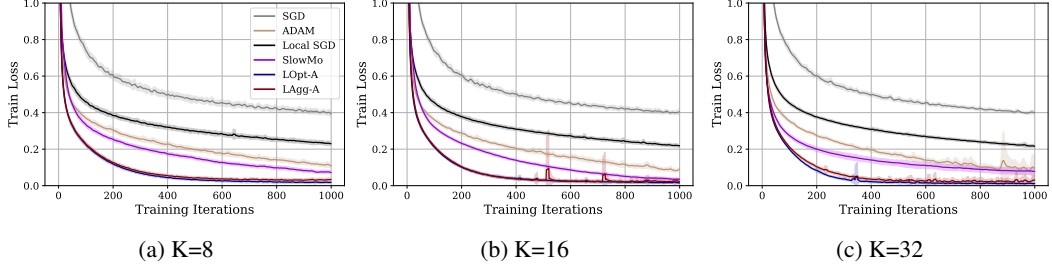

(a) K=8        (b) K=16        (c) K=32

Figure 3: **LAgg-A outperforms all optimizers for** $K \in \{8, 16, 32\}$ **workers**. All training curves are reported for the $28 \times 28$ FMNST dataset. The top row plots training curves for a small CNN, while the bottom row plots training curves for an MLP. All experiments use $H = 4$.

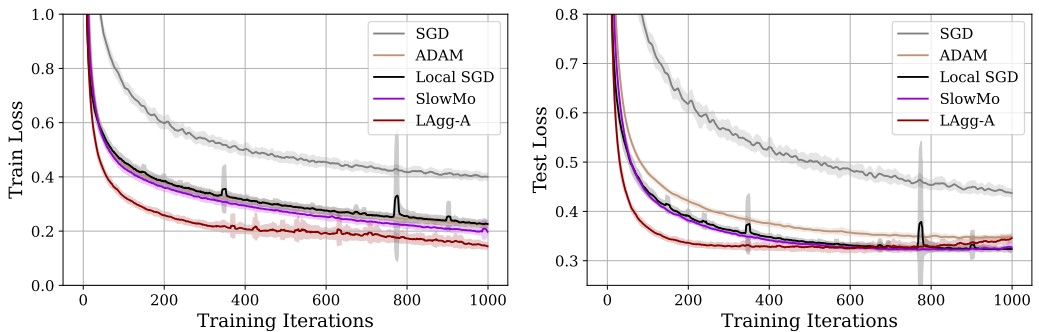

Figure 4: **Directly targeting validation loss during meta-training obtains strong performance on the test set. (Left)** plots the training loss when optimizing models on the 2-Layer MLP FMNIST task, while **(Right)** plots the validation loss. We observe that learned optimizers trained to optimize validation loss generalize in our setting. All models were trained with $K = 8$ and $H = 4$. Hand-designed optimizers were hyper-parameter-tuned to the validation set, while LAgg-A was meta-trained to optimize validation loss.

**Communication-Efficient Distributed Baselines**    We provide two communication-efficient distributed baselines: local SGD (Stich, 2019) and SlowMo (Wang et al., 2019). An extensive hyper-parameter search is conducted for each baseline in every configuration. We detail the search process and report the best hyperparameters in section C of the supplement.

## 4.2   EVALUATING LAGG-A AND LOPT-A IN DISTRIBUTION

In this section, we evaluate our proposed optimizers on three datasets using $H = 4$ iterations and $K = 8$ workers. Following the evaluation protocol of Metz et al. (2022a), in each case, we meta-train on a task (dataset and architecture pair) and perform evaluation on a new seed. That is, in distribution evaluations test the generalization of the optimizer to new initialization of the model and new ordering of the data. Results on FMNIST 2-Layer MLP (left), CIFAR-10 CNN (center), and ImageNet 3-Layer MLP (right) are reported in Figure 1. We observe that our learned optimizers enjoy strong convergence, obtaining lower training loss in fewer iterations than all baseline models. Note that the SlowMo is well-tuned and represents a very competitive approach in the class of methods that perform local updates Wang et al. (2019).

## 4.3   THE EFFECT OF LOCAL ITERATIONS ($H$)

We now analyze our local learned optimizers' capability to scale to a larger number of local iterations ($H$). Specifically, we vary $H \in \{4, 8, 16\}$ and meta-train our learned optimizers on the FMNIST 2-Layer MLP task for each case. We also report the performance of corresponding tuned baselines with the equivalent batch size. The results are reported in Figure 2. We also show the number of

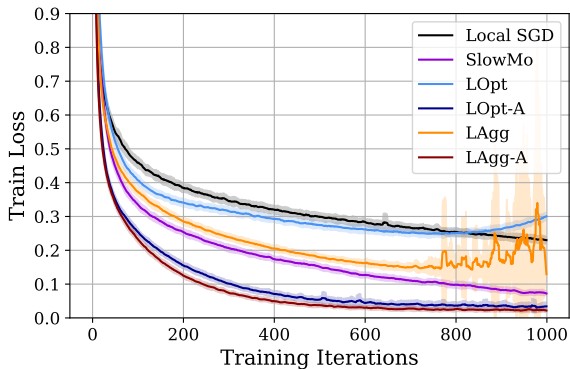

Figure 5: **Effect of different features on optimizer performance.** Each curve is an average over 10 trials with different seeds. Shaded regions represent one standard error from the mean. Each learned optimizer is trained and tested on FMNIST 2-Layer MLP at $H = 4$ and $K = 8$. Baseline models use tuned hyper-parameters reported in Table 3 of the supplement.

communications to achieve a fixed training loss (0.2) in Table 1. We observe that even for relatively high $H$ (Lin et al., 2018) there is an improvement over the strong communication-efficient baselines. As expected, Table 1 illustrates higher $H$ yields more rapid convergence on a per training iteration basis (due to more samples being processed). We also observe that LAgg-A begins to show a substantial advantage compared to LOpt-A at this higher $H$ value. We thus focus on this optimizer in our subsequent meta-generalization studies.

### 4.4    EFFECT OF THE NUMBER OF WORKERS ($K$)

In Figure 3 we evaluate the performance of our method as the number of workers ($K$) increases. Similarly to section 4.3, we vary $K \in \{8, 16, 32\}$ and meta-train our learned optimizers on the FMNIST 2-Layer MLP task for each case. We observe that our local learned optimizers gracefully scale to more workers, reaching a lower loss in fewer iterations than all baselines by a significant margin in each case.

### 4.5    ABLATING ADAFACTOR FEATURES

Our learned optimizer leverages powerful per-parameter learned optimization features proposed in Metz et al. (2022a). Here we investigate how important these are to the performance of the optimizer. Specifically, we consider directly feeding the $\Delta_t$ or $\Delta_t^{1..K}$ to the learned optimization MLP network along with the parameter value and the 11 time features without adding any of the other momentum or AdaFactor features described in supplement A. We denote these baselines as LOpt and LAgg, respectively (excluding the -A). Results are presented in Figure 5. We observe that a large improvement in convergence and training stability is obtained by using AdaFactor features in both cases. However, we note that the performance of LOpt and LAgg alone still experiences improved convergence early in training with respect to local SGD. These baselines have no momentum calculations and the optimizer is an MLP (as opposed to a recurrent model) thus there is no way to maintain history information (unlike SlowMo's momentum). It is therefore notable that LAgg can achieve similar, albeit slower, convergence to SlowMo during the first 600 iterations. However, LAgg does seem to cause training instability from iteration 800 onwards. Interestingly, the models trained with AdaFactor features do not suffer from such instabilities, despite being trained with the same schedule as LAgg, further demonstrating their benefit.

### 4.6    ABLATING OUTER LOOP GENERALIZATION

Following conventions in the learned optimization literature (Metz et al., 2022b;a) our focus in this work has been demonstrating the efficient convergence of the learned optimizer. Thus in our experiments, the outer loop of the meta-learning problem (Eq. equation 1) evaluates the training data. In this section, we demonstrate that we can also obtain strong performance on the validation data using our learned optimizer. Figure 4 plots the training loss (left) and test loss (right) of our local learned optimizers trained using the validation loss objective and baselines tuned using validation

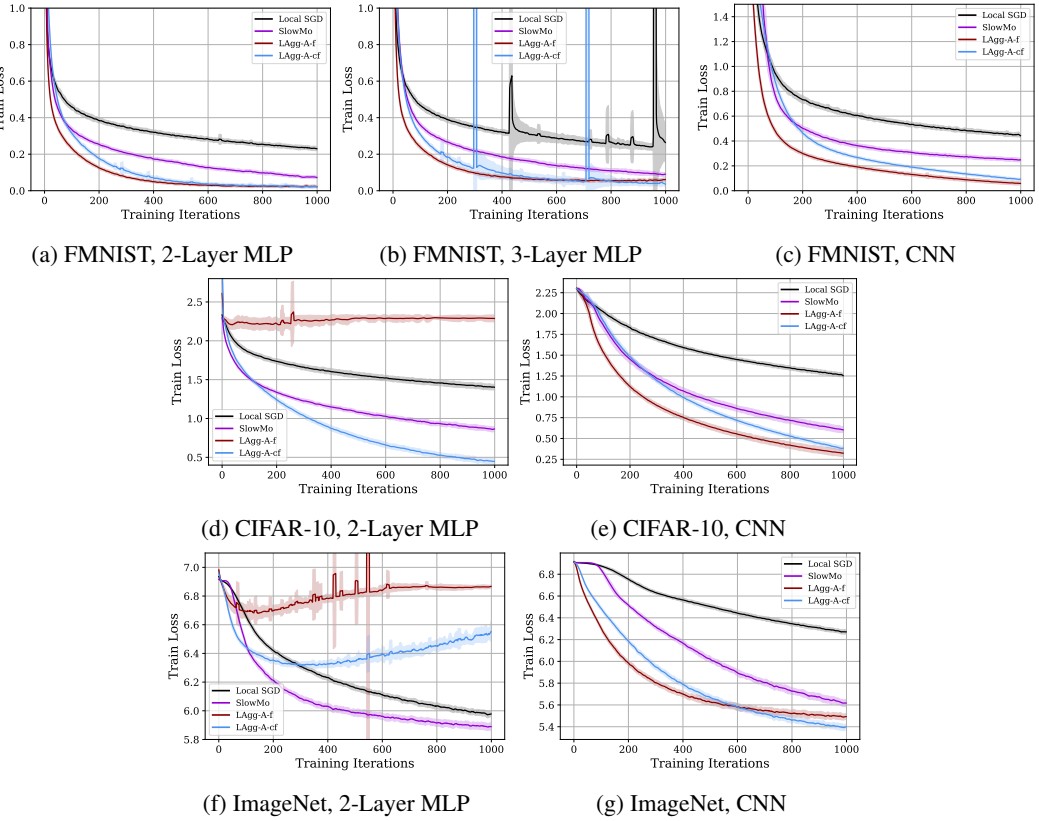

Figure 6: **Meta-generalization to new datasets and new architectures**. All optimizers were meta-trained and hyper-parameter tuned for task (a). Meta-generalization is evaluated in three progressively more difficult settings: new architectures same dataset (plots (b),(c)), new dataset same architecture (plots (d),(f)), and new dataset and new architecture (plots (e),(g)). Local learned optimizers achieve strong generalization to different architectures on the same dataset, but experience difficulties optimizing the same architecture on a new dataset. However, the improvements of performance from LAgg-A-f to LAgg-A-cf in plot (f) shows that these issues can be mitigated by scaling training tasks. Finally, both local learned optimizers evaluated generalize outside of the training data distribution and architecture in plots (e) and (g).

loss. We observe a similar trend with respect to training loss for LAgg-A: it improves convergence across the board when compared to baseline models. On the test loss plot, LAgg-A converges significantly faster than other baselines reaching a test loss around iteration 200 that baselines only reach after 600 iterations of training. We believe this strength of LAgg-A is attributable to the meta-training objective (eq 1) that weights the validation loss from any iteration equally, encouraging LAgg-A to immediately decrease the loss.

### 4.7 META-GENERALIZATION

This section evaluates the meta-generalization capabilities of our locally learned optimizers in communication-efficient settings. The results are reported in Figures 6 and 7. In Figure 6, we evaluate generalization in three progressively more difficult settings: new architectures same dataset (plots (b),(c)), new dataset same architecture (plots (d),(f)), and new dataset and new architecture (plots (e),(g)). In Figure 7, we evaluate the capability of our local learned optimizers trained at one H value to generalize to another. For simplicity, we focus on evaluating LAgg-A as it generally performs as well or better than LOpt-A. **LAgg-A-f** is trained on the FMNIST, 2-Layer MLP task, while **LAgg-A-cf** is trained on a two-dataset task using FMNIST and CIFAR-10 with the same 2-Layer MLP. All baseline models use hyperparameters tuned on the FMNIST 2-Layer MLP task.

Every model is trained using $K = 8$ and $H = 4$ with the exception of **LAgg-A H=16** (trained using $K = 8$ and $H = 16$).

**Generalization to unseen architectures**  We observe that our learned optimizers can generalize to unseen architectures (Fig 6 plots (b),(c)). In particular, LAgg-A-f trained on 2-Layer MLP tasks can perform well on a CNN and an MLP of different depth, highlighting the practicality of our approach. Performance in the case of the CNN is particularly strong without having observed this architecture during training

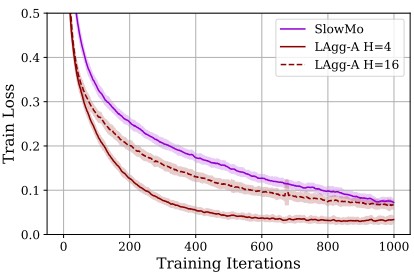

Figure 7: **LAgg-A trained at $H = 16$ generalizes to $H = 4$.** We observe that **LAgg-A H=16** trained at $H = 16, K = 8$ improves upon a strong SlowMo baseline at $H = 4, K = 8$.

**Generalization to unseen datasets**  We observe that LAgg-A meta-trained on FMNIST 2-Layer MLP struggles to optimize the same architecture on CIFAR-10 and Imagenet. We note, however, that including an additional task (CIFAR-10, MLP) during meta-learning can significantly improve performance. Specifically, we observe that this learned optimizer (LAgg-A-cf) is able to generalize to both of its in-distribution tasks (CIFAR-10 and FMNIST MLP) as well as improve performance on Imagenet MLP. This suggests that stronger meta-generalization can be achieved by scaling the training tasks in our communication-efficient setting as has been demonstrated for standard optimization settings in the learned optimization literature (Metz et al., 2022b).

**Generalization to unseen datasets and architectures**  Interestingly, we observe (Fig 6 plots (e),(g)) that both learned optimizers, Lagg-A-f and LAgg-A-cf achieve strong generalization when varying both the dataset (CIFAR-10 and ImageNet) and the architecture (CNN).

**L-Agg-A trained for more local updates can generalize**  We also evaluate whether models trained with a given number of local steps can generalize to variations in the number of local steps. Results are shown in Figure 7, here we observe that a model trained with H=16 can still perform competitively (exceeding SlowMo) when meta-evaluated with 4 local steps.

These results establish the existence of meta-generalization capabilities (Metz et al., 2022a;b) for locally learned optimizers. Moreover, they demonstrate that such optimizers can also generalize to different values of $H$, suggesting that it is possible to obtain local learned optimizers that are general in $H$ and in tasks by scaling training compute and task variety while using higher $H$ values.

## 5 CONCLUSION

We have demonstrated the utility of learned optimization for improving communication-efficient distributed training of deep networks. We have proposed two local learned optimizer architectures for this setting: LAgg-A and LOpt-A. Our results illustrate that these optimizers can effectively be applied in communication-efficient distributed settings; that they can scale to larger values of $H$ and $K$; that local learned optimizers exhibit generalization capabilities to unseen architecture and datasets; and that they are also capable of generalizing from large to smaller $H$ values.

These findings establish learned optimization as a promising direction for improving communication-efficient distributed training algorithms for deep learning. Given the generalization capabilities of local learned optimizers, future work focusing on scaling such approaches to larger number of architectures, datasets, and local steps can potentially obtain very efficient training algorithms. Furthermore, applications of these techniques can be made in other communication-efficient distributed learning contexts, specifically decentralized and federated learning.

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

## A  LEARNED OPTIMIZERS ARCHITECTURE AND FEATURES

Both our proposed learned optimizers, LOpt-A and LAgg-A, consist of a 2 hidden layer, 32 hidden nodes per layer MLP with a ReLU activation function. They share some common input features that are detailed in Table 2 and denoted $\text{ADA}(\cdot)$ in the main text. All but the time features are normalized to have a second moment of 1 across the tensor. These features track momentum, second moments and AdaFactor row and column features stored in the inner state $\boldsymbol{u}_t$ of the learned optimizer, which are updated at each time step. Unlike prior work, all our computations are based on the average update, $\Delta_t$. All the coefficients, $\beta_i$, are learnable parameters obtained during meta-optimization. More details of the computations for obtaining the AdaFactor row and column features can be found in Shazeer & Stern (2018).

Table 2: **Common input features of LOpt-A and LAgg-A.**

| Description | |
| --- | --- |
| parameter value | $w_t$ |
| 3 momentum values with coefficients $\beta_1, \beta_2, \beta_3$ | $m_{t,i} = \beta_i m_{t-1,i} + (1 - \beta_i)\Delta_t$ |
| second moment value computed from $\Delta_t$ with decay $\beta_4$ | $v_t = \beta_4 v_{t-1} + (1 - \beta_4)\Delta_t^2$ |
| 3 values consisting of the three momentum values normalized by the square root of the second moment | $\frac{m_{t,i}}{\sqrt{v}}$ |
| the reciprocal square root of the second moment value | $\frac{1}{\sqrt{v}}$ |
| 3 $\Delta_t$ AdaFactor normalized values | $\Delta_t \times$ ROW FACTOR $\times$ COLUMN FACTOR |
| 3 tiled AdaFactor row features with coefficients $\beta_5, \beta_6, \beta_7$, computed from $\Delta_t$ | $r_{t,i} = \beta_i r_{t-1,i} + (1 - \beta_i)\text{ROW\_MEAN}(\Delta_t^2)$ |
| 3 tiled AdaFactor column feature with coefficients $\beta_5, \beta_6, \beta_7$ computed from $\Delta_t$ | $c_{t,i} = \beta_i c_{t-1,i} + (1 - \beta_i)\text{COL\_MEAN}(\Delta_t^2)$ |
| the reciprocal square root of the previous 6 features | $\frac{1}{\sqrt{r_{t,i} \text{ OR } c_{t,i}}}$ |
| 3 $m$ AdaFactor normalized values | $m_{t,i} \times$ ROW FACTOR $\times$ COLUMN FACTOR |
| 11 time features computed from the current timestep $t$ and $x \in \{1, 3, 10, 30, 100, 300, 1000, 3000, \text{10k}, \text{30k}, \text{100k}\}$ | $\tanh\left(\frac{t}{x}\right)$ |

The state tracked by the learned optimizer thus includes

$$\boldsymbol{u}_t = \{m_{t,1}, m_{t,2}, m_{t,3}, v_t, r_{t,1}, r_{t,2}, r_{t,3}, c_{t,1}, c_{t,2}, c_{t,3}, t\}$$

Note that the AdaFactor row features are computed on a per-tensor basis. Specifically, the ROW\_MEAN and COL\_MEAN operation is applied on a per tensor basis. For each tensor, the corresponding components of $\Delta_t^2$ are reshaped and their row and column means are computed. For more details please see Shazeer & Stern (2018).

Our first learned optimizer, LOpt-A, has another input feature, $\Delta_t$, the average of all $\Delta_t^{(k)}$ coming from the $K$ workers, for a total of 39 input features. Our second learned optimizer, LAgg-A, had $K$ other input features which are all the different $\Delta_t^{(k)}$ coming from the $K$ workers, for a total of $38 + K$ input features. Those features are also normalized like the others.

Both MLP output two values, a magnitude $m$ and a scalar direction $d$ that are use to compute the parameter update with the formula $\lambda_1 d \exp(\lambda_2 m)$ where $\lambda_1$ and $\lambda_2$ are constants values of 0.001 to keep initial step sizes small.

With all of this in mind we can compute the number of meta-parameters $\phi$ in the MLP for each of our learned optimizers. LOpt-A has a total of 2402 meta-parameters, while LAgg-A for values $K \in \{8, 16, 32\}$ respectively have 2626, 2882 and 3394.

## B  META-TRAINING PROCESS

As stated in equation 1, our meta-learning objective is the average loss over $T$ iterations. This optimization problem usually requires long unrolls of the compute graph. We alleviate problems that can arise from long unrolls by using Persistent Evolution Strategies (PES) to compute estimates of the gradients. In our study, we use a truncation schedule that samples unroll lenghts $N$ from a log-uniform distribution with a minimal value of $N = 100$ and a maximum value of $N = 1000$ (the maximum value with which we evaluate our learned optimizers). The idea being that we don't always need to compute the whole inner problem each time and we can rather use information from a shorter subsequence of the problem to update the weights $\phi$ of our learned optimizer. Our partial unrolls used with PES have a length of 50.

For most of the learned optimizers in our study, we meta-trained for 5 000 steps. The only exceptions are the learned optimizers used in section 4.3 and the learned optimizer meta-trained for ImageNet that were meta-trained for 10 000 steps. During meta-training, we used AdamW as our optimizer with a warmup cosine decay schedule. The learning rate starts at $3e-10$ and warms up linearly to the peak value of $3e-3$. It then decays to the final value of $1e-3$ until the end of meta-training.

## C  BASELINES

For every configuration in which we used the baseline optimizers, namely the architecture, the dataset and the different values of $K$ and $H$, we ran an exhaustive hyperparameter sweep over the following values. For SGD and Adam, we searched over the learning rate $\alpha \in \{1, 5e-1, 1e-1, 5e-2, 1e-2, 5e-3, 1e-3, 5e-4, 1e-4, 5e-5, 1e-5\}$. For local SGD, we searched over the local learning rate $\gamma \in \{1, .5, .3, .1\}$. For SlowMo, we varied the local learning rate $\gamma \in \{1, 0.5, 0.3, 0.1\}$, the slow learning rate $\alpha \in \{1/\gamma, 5e-1/\gamma, 1e-1/\gamma, 5e-2/\gamma, 1e-2/\gamma, 5e-3/\gamma, 1e-3/\gamma, 5e-4/\gamma, 1e-4/\gamma, 5e-5/\gamma, 1e-5/\gamma\}$ and the momentum $\beta \in \{0.99, 0.95, 0.9, 0.85, 0.8, 0.75, 0.7, 0.65, 0.6, 0.55, 0.5\}$. The best hyperparameters for each configuration are regrouped in Table 3.

Table 3: **Best hyperparameters for baselines**

| Configuration | SGD ($\alpha$) | Adam ($\alpha$) | local SGD ($\gamma$) | SlowMo ($\gamma$ / $\alpha$ / $\beta$) |
|---|---|---|---|---|
| FMNIST $28 \times 28$, 2-Layer MLP, $K = 8, H = 4$ | 0.1 | 0.01 | 0.3 | 0.1 / 1 / 0.95 |
| FMNIST $28 \times 28$, 2-Layer MLP, $K = 8, H = 8$ | 0.1 | 0.005 | 0.3 | 0.1 / 1 / 0.95 |
| FMNIST $28 \times 28$, 2-Layer MLP, $K = 8, H = 16$ | 0.1 | 0.005 | 0.1 | 0.1 / 1 / 0.95 |
| FMNIST $28 \times 28$, 2-Layer MLP, $K = 16, H = 4$ | 0.1 | 0.005 | 0.5 | 0.1 / 1 / 0.95 |
| FMNIST $28 \times 28$, 2-Layer MLP, $K = 32, H = 4$ | 0.1 | 0.005 | 0.5 | 0.3 / 1.66 / 0.9 |
| CIFAR-10 $32 \times 32$, CNN, $K = 8, H = 4$ | 1 | 0.01 | 1 | 0.5 / 2 / 0.9 |
| ImageNet $32 \times 32$, 3-Layer MLP, $K = 8, H = 4$ | 1 | 0.001 | 0.3 | 0.1 / 1 / 0.85 |

