# OpenReview forum: "Learning Communication-Efficient Optimizers"
_ICLR.cc/2024/Conference — ICLR 2024 Conference Withdrawn Submission_

### Official Review · Reviewer_eXvU · 2023-10-20

**Soundness:** 2 fair
**Presentation:** 1 poor
**Contribution:** 2 fair
**Rating:** 3
**Confidence:** 3

**Summary:**

This paper leverages the learned optimization framework to enhance Local SGD, seeking for a more efficient local gradient algorithm. Specifically, the proposed method views the updates within one communication round as pseudo-gradients. The learned optimizer is at the server side and takes in functions of these pseudo-gradients as input. Experimental results show that the learned communication-efficient optimizers can lead to faster convergence of the training loss, when compared with Local SGD, SlowMo, SGD and Adam, given the same number of communication rounds. Additionally, the paper argues that the learned optimizers can generalize to new datasets and architectures.

**Strengths:**

Embracing the "learning-to-learn" strategy is commendable for crafting more communication-efficient methods.  Communication can become the bottleneck for both large-scale datacenter distributed training and federated learning. Strategies to reduce the communication overhead like performing local updates and compression usually come at the cost of slower convergence. Instead of mannually designing better optimizers, which demands a good knowledge of optimization theory, using the meta-learning approach to learn the optimizer can be a promising solution.

**Weaknesses:**

Major:

1. **The experiments are conducted on toy MLPs and CNNs, which cannot convince me of the usability of the proposed methods**. Communication-efficient optimizers are typically deployed in two main contexts: datacenter distributed training and federated learning. In the former, the scale of the models leads to communication lags due to the vast volume of data exchanged. In the latter, although the models might not be that large, optimization is complicated by factors such as data heterogeneity and partial client participation. The experiments in this study do not resonate with the complexities of either setting, even without testing on a small ResNet. When using such toy models, the need for communication-efficient optimizers becomes questionable.
2. **Insufficient Insight into the Methodology**. Even if we assume that the proposed method performs well in practical settings, this paper does not provide insight into its success. Given that the learned optimizer can perform complex operations and that local updates add to the complexity., it's crucial to delve deeper and find out the key contributing factors.

3. **The authors are urged to improve their writing.** I struggled to grasp the authors' intentions during reading. For example, the description of the proposed methodology in Section 3 is unclear. The text preceding Algorithm 1 does not reference the 'adafactor feature' $A_t$, yet it unexpectedly appears within the algorithm. This oversight can lead to confusion. The terms 'adafactor feature' and 'learned optimizer state' are also vague, requiring more illustration. The loss summation in eq (1) references the step $t$, but its absence in the summand is perplexing. In the experimental section, the x-axis labels of the figures should more accurately read 'the number of communication rounds' rather than 'training iterations' .

Minor:

1. Missing reference:  Lv, Kaifeng, Shunhua Jiang, and Jian Li. "Learning gradient descent: Better generalization and longer horizons." *International Conference on Machine Learning*. PMLR, 2017.

2. Grammatical mistakes/typos:

   - Section 1, lines 3-4: "the communication overhead associated with distributed SGD can often lead to inefficient use of computing resources and *reduced* wall clock times (Lin et al., 2018)". reduced->increased
   - Section 3, line 3: "We take H local step of SGD", step->steps

   The authors should go through their paper carefully to correct all such mistakes.

**Questions:**

1. It is weird that the learned optimizer underperforms when the architecture remains the same but the dataset changes but performs well when both change. Can the authors explain more?

2. Can optimizers learned on smaller values of $H$ generalize to a larger $H$?

---

### Official Review · Reviewer_WpRV · 2023-10-29

**Soundness:** 3 good
**Presentation:** 3 good
**Contribution:** 2 fair
**Rating:** 5
**Confidence:** 3

**Summary:**

This work proposes to apply meta-learning on local SGD. Experimental results show that the proposed learned optimizers can outperform local SGD in the image classification task.

**Strengths:**

1. This work's presentation is good and the work is easy to understand
2. A detailed experimental comparison between this work and previous works is presented.

**Weaknesses:**

1. Regarding the chosen architecture in the experiments: the studies employed basic models such as MLP and small-scale CNNs as their neural architectures. I think these models are overly simplistic, raising doubts about the applicability of the proposed method to more complex, widely discussed architectures like ResNet and Transformer-based models.

2. Concerning the generalization capability of the proposed optimizer: while it is asserted in the abstract that the optimizer shows generalization, it often falls short in this aspect. For example, the optimizer diverges in figures 6d and 6f. This raises concerns about the method's practicality since retraining the optimizer for each new problem is time-consuming.

3. The presentation of experimental outcomes is somewhat unclear. Specifically, in figures 1–5, the x-axis represents training iterations, which equals the number of communication iterations. This seems unfair to SGD and ADAM, as algorithms based on local SGD undertake more updates per communication. It would be more appropriate to display the count of local updates on the x-axis. Moreover, in real-world scenarios, communication time often overlaps with computation, making it less time-consuming. Hence, for most neural network structures, local updates remain the primary bottleneck.

4. In figures 2, 5, and 6, the proposed optimizer sometimes exhibits divergence. Incorporating a trust-region check to ensure convergence of the optimizer would be beneficial.

**Questions:**

n/a

---

### Official Review · Reviewer_jUos · 2023-10-30

**Soundness:** 3 good
**Presentation:** 3 good
**Contribution:** 2 fair
**Rating:** 5
**Confidence:** 4

**Summary:**

This paper proposed and evaluated two methods on learning communication efficient optimizers based on the local SGD framework. Experiments show that the proposed method can beat traditional methods in shallow network training tasks, and that the trained optimizers generalize well to different settings.

**Strengths:**

1.	The learned optimizers utilize simple information gathered from workers, and generalize well to different network structures and datasets.
2.	The number of parameters to train are relatively small.
3.	The learned optimizers are general-purpose in the sense that it may generalize to problems with different dimension.

**Weaknesses:**

1.	It is unclear whether the trained optimizers can generalize to different local update hyperparameters other than the number of inner-loop iterations $H$, e.g. the local learning rate $\gamma$, the local batch size $\mathcal{B}_{loc}$.
2.	The optimizer learned by LAgg-A cannot generalize to settings with different number of workers, because the input size $K+38$ is correlated with the number of workers $K$.
3.	The train/test problems for the learned optimizers are mainly shallow networks on image classification tasks. The lack of diversity in test problems may not justify well on the generalization properties of learned optimizers.

**Questions:**

1.	It seems that the learned optimizers are unaware of the local learning rate $\gamma$, how is this $\gamma$ set in each experiment? Please clarify.
2.	What does it mean by *local learned optimizers*? Can the optimizers be learned locally on each worker, or can it be used locally during inference time? If the term *local* is only concerned with the local SGD framework that your methods are based on, it’s better to use another term instead to avoid ambiguity.
3.	As mentioned in the weakness, it seems that the learned LAgg-A optimizer cannot generalize to settings with a different number of workers. If this is true, the statement made in subsection 4.4 *our local learned optimizers gracefully scale to more workers* may have ambiguity, since the evaluated optimizers are actually different ones trained separately. Otherwise, please clarify how a learned MLP with input dimension $K+38$ is generalized to input size with larger $K$’s.
4.	Similar to the previous point, it’s unclear whether the curves of learned optimizers are of the same trained weight, or they are trained separately.
5.	It seems that none of the baseline methods are learned optimizers. Is it because no learning-based algorithms are suitable for comparison?

---

### Official Review · Reviewer_QnQe · 2023-11-01

**Soundness:** 2 fair
**Presentation:** 3 good
**Contribution:** 2 fair
**Rating:** 3
**Confidence:** 4

**Summary:**

This article introduces LAgg-A, a communication-efficient optimizer designed to improve the aggregation of local updates from multiple clients. LAgg-A utilizes these local updates as input features for a learning optimizer MLP, enabling it to learn effective aggregation strategies on the server side. The study demonstrates the superiority of the learned optimizer over local SGD and its variants across diverse tasks and highlights its modest generalization capabilities.

**Strengths:**

The expriemental resulsts indicates this learning optimizer is effective for training simple neural networks.

**Weaknesses:**

- The practicality of LAgg-A is called into question, particularly in the context of modern neural networks with numerous layers and millions/billions of parameters. The experimental application of LAgg-A is limited to simple 2/3-layer networks with thousands of parameters. Moreover, LAgg-A appears to be an architecture-specific optimizer although it shows some generalization abilities. This restricts its usability compared to widely applicable optimizers like SGD and Adam.

- The novelty of LAgg-A appears to be incremental, as it extends the small_fc_lopt optimizer proposed in [1] for distributed learning. The main innovation seems to involve substituting the first/second gradient momentum in [1] with local updates from multiple clients as input features for the learning optimizer MLP. A more substantial leap in innovation would be beneficial.

**Questions:**

N/A